# COVID-19 in Children: Molecular Profile and Pathological Features

**DOI:** 10.3390/ijms242316750

**Published:** 2023-11-25

**Authors:** Ruslan A. Nasyrov, Dmitry O. Ivanov, Olga L. Krasnogorskaya, Vladimir N. Timchenko, Elena P. Fedotova, Alexander S. Chepelev, Veronika A. Galichina, Nadezhda A. Sidorova, Nikolai M. Anichkov

**Affiliations:** The Prof. D.D. Lohov Department of Pathological Anatomy with Course of Forensic Medicine, Saint Petersburg State Pediatric Medical University Ministry of Public Health Care of the Russian Federation, St. Litovskaya, 2, 194100 St. Petersburg, Russia; doivanov@yandex.ru (D.O.I.); krasnogorskaya@yandex.ru (O.L.K.); timchenko220853@yandex.ru (V.N.T.); kris6060@mail.ru (E.P.F.); spmmed@yandex.ru (A.S.C.); galichina-nika@mail.ru (V.A.G.); nadyukochka@mail.ru (N.A.S.); anichkov@bk.ru (N.M.A.)

**Keywords:** COVID-19, children, lungs, morphology, immunohistochemistry, apoptosis

## Abstract

Although the World Health Organization has declared the end of the COVID-19 pandemic, doctors continue to register new cases of the disease among both adults and children. Unfortunately, the course of COVID-19 in children can have a severe form, with death being a potential outcome. The absence of published works discussing the pathological morphology of COVID-19 in children prevents the objective analysis of the disease’s pathogenesis, including among the adult population. In this vein, the objective of our study is to identify the morphological features of the lungs’ involvement and evaluate virus–host interactions in the case of COVID-19 in patients at a pediatric medical practice. We present the results of the study of the lungs of three children who died due to COVID-19, highlighting the predominant involvement of their respiratory organs at different stages of the disease (5, 21, and 50 days). This article presents data obtained from histopathological and immunohistochemical investigations, taking into account the results of clinical and laboratory indicators and intravital and postmortem SARS-CoV-2 PCR investigations. The common finding of all of the examined COVID-19 cases is the involvement of the endothelium in microcirculation vessels, which are considered to be a primary target of various pathogenic influencing factors. We also discuss both the significance of apoptosis as a result of virus–host interactions and the most likely cause of endothelium cell destruction. The results of this study could be useful for the development of endothelium-protective therapy to prevent the progression of disseminated intravascular coagulation syndrome.

## 1. Introduction

The COVID-19 pandemic caused an unprecedented situation in various spheres of the public health service. Despite the end of the pandemic being declared, functioning treatment management services, and existing controlled vaccination programs, new cases of COVID-19 continue to be registered [1,2]. The infection mostly affects the adult population, while children are affected less often [3]. Only 3.6% of the total number of identified COVID-19 cases are child patients, and just 0.6–2.0% of them are admitted to an intensive care unit [4]. Nevertheless, clinical practice indicates that COVID-19 can have a severe form in children that can lead to death [5]. Children with a comorbid pathology, in particular those with oncological diseases, belong to a particularly high-risk group, as they often have an extremely severe course of coronavirus infection and, thus, a higher risk of death [6].

It has become an axiom that postmortem investigations (autopsies) are a “gold standard” means of determining a disease’s pathogenesis [7,8]. Nowadays, despite a large number of publications being devoted to postmortem investigations of the lungs in cases of COVID-19 in adults [9,10,11,12,13,14,15,16,17,18], the understanding of the pathological morphology of this disease based on the results of immunohistochemical investigations is still limited. The data regarding the study of children who died due to COVID-19 infection are of extremely poor quality [5]. Indeed, the majority of suggestions concerning the pathogenesis of COVID-19 in children are directly connected to the absence of concrete facts about the pathological morphology of the disease. The assessment of virus–host interactions is greatly important in terms of understanding the infection’s course and development [19]. It is necessary to note that the investigation of COVID-19 pathogenesis is also the basis for understanding another topical problem: the development of post-COVID conditions.

The aim of our study is to identify the morphological features of the lungs’ involvement and evaluate virus–host interactions in cases of COVID-19 in patients treated at a pediatric medical practice.

## 2. Results

Patient 1. The duration of the disease was 5–6 days. The epithelium was cast off in most parts of the bronchial tubes. In the majority of the alveoli lumen (70%), there was an interalveolar hypostasis: serous or serous-hemorrhagic exudate. In some alveoli (5%), we identified tape-like and slightly stained eosinophilic films covering the inside part of the alveoli walls (Appendix A). Interalveolar septa were unevenly thickened from infiltration by lymphocytes (up to 10 CD8+ cells in the field of vision under ×400 magnification) and some visible macrophages. The vessels of interalveolar septa were sharply full-blooded, and there were marked epithelium desquamation and mixed blood clots in their lumina. Moreover, all studied tissue areas of the lungs had massive, widespread hemorrhages (Figure 1A).

SARS-CoV-2 was found in the material of the trachea and both lungs during the postmortem PCR investigation. According to the data collected during the immunohistochemical investigation, there was an acute expression of SARS-CoV-2 antigens (nucleocapsid) in the bronchial epithelium, with antigens being expressed (++) on the walls of the arterioles and in venules (++), while positive expression (+) was noted in the capillaries of the interalveolar septa (Figure 1B).

The capillary network was found to be irregular, and its fragmentation and deendothelialization were noticed during CD31 immunohistochemical staining (Figure 1C). According to the data collected during the immunohistochemical investigation, the acute expression of CD95 was identified in the epithelium of the bronchi and the endothelia of the micro-vessels (1D), macrophages (+++), and a small number of apoptosis particles located in the interalveolar septa (+).

Patient 2. The duration of the disease was 20 days. The light microscopy identified a total decrease in airiness in all lobes of the lung. The cast-off epithelium was replaced by a thickened basic membrane covered by a fibrin fiber with disseminated lymphocyte infiltration. The widespread foci of inflammatory infiltration were noticed in the surrounding tissues and the walls of the bronchi, with CD8+ lymphocytes predominantly being present (Appendix A). We revealed a subtotal (95%) formation of dense protein aggregates in the alveoli of all lung tissues (Figure 2A) and tape-like fibrin masses (Appendix A). The interalveolar septa were thickened, and they contained lymphocytes, apoptosis particles, and fibroblasts. The stasis of red blood cells and hybrid clots could be seen in the lumina of the capillaries of the interalveolar septa. SARS-CoV-2 was found in the material of the trachea and both lungs during the postmortem PCR investigation. According to the immunohistochemical investigation, the expression of SARS-CoV-2 antigens in the epithelium of the bronchi was acute (+++) and sub-totally spread in nature (Figure 2B). The acute expression of SARS-CoV-2 antigens in blood vessel endotheliocytes (++) could be observed (Figure 2C), and a marked expression of SARS-CoV-2 (++) was identified in the capillaries of the interalveolar septa. Furthermore, antigen expression was observed in the monocytes of the vessel lumen (+) and alveolar macrophages (+). During the immunohistochemical investigation, an irregular intensity of CD31 expression was noted, and the majority of endothelium cells in the vessels in microcirculation lacked accurate identification (Appendix A). An acute positive expression of the apoptosis marker CD95 (+++) was revealed in the bronchial epithelium, the endothelium of vessels of different sizes, the interalveolar septa, and the apoptosis particles (Figure 2D) and alveolar macrophages (Appendix A).

Patient 3. The duration of the disease was 50 days. The tissue airiness was lowered by 95% in the segments of both lungs. There was a disseminated lymphocytic infiltration in the walls of small- and middle-sized bronchi. Fibrinogenous masses were identified on the erosive surfaces of the mucous membranes of the bronchi and their lumina. Fibrinogenous masses with admixtures of cast-off alveolar macrophages and red blood cells, which completely filled the whole volume of the alveoli, were visible (Appendix A). Tape-like fibrin masses covering the alveoli were also distinctly seen in cases of staining via trichrome according to Masson’s method (Figure 3A). Moreover, we identified the areas of lung tissue that experienced intra-alveolar hemorrhage. Interalveolar septa were slightly thickened and infiltrated by disseminated lymphocytes and macrophages with large hyperchromic nuclei. The stasis of red blood cells could be seen in the capillaries of the interalveolar septa, with hybrid clots in some vessels. According to the intravital PCR investigations, SARS-CoV-2 was found in the material taken from the oropharynx and the nose, as well as in the broncho-alveolar lavage at days 31, 38, 43, and 47 of the disease, but SARS-CoV-2 was not found during the postmortem PCR investigations of the material taken from the trachea, the main bronchi, and the tissue of both lungs. The SARS-CoV-2 antigen was identified in the form of a small number of blocks of different sizes (3–5 in each field of vision with ×400 magnification) during the immunohistochemical investigation, and it appeared to be disseminated in the lung tissue (Figure 3B). The identification of its exact localization, namely in which cells it was located, was difficult to perform. Presumably, these were most likely macrophages where a high concentration of COVID-19 antigen was previously identified. The endothelial cells of the capillary network were not identified in the lung tissue in the case of CD31 staining, and they could only be seen as adelphomorphous and slightly stained and as small structures (Figure 3C).

The expression of the apoptosis marker CD95 noted in some endothelial cells found in the arterioles, venules, and interalveolar septa, as well as in the epithelia of bronchi and the apoptosis particles was identified as being weak positive/doubtful (+) in nature (Figure 3D).

## 3. Discussion

Our study presents an analysis of the autopsy materials derived from three cases of COVID-19 in children. Histological changes in the lungs were investigated in the cases in which the disease had various durations, with the use of immunohistochemical and PCR investigations allowing us to perform a pathogenetic evaluation of the different stages of the disease’s development. Therefore, Patient 1, for whom the duration of the disease was 5 days, experienced changes characteristic of the early period of the exudative phase of diffuse alveolar damage. In another case, Patient 2 had significant changes on the 20th day after the disease’s onset: 95% of the alveoli were filled with a dense, serofibrinous exudate, and there were hybrid clots in the vessels of different sizes in all fields of vision. In contrast to the previous case, there was significant T-lymphocytic inflammatory infiltration in the interalveolar septa, and everywhere else, we noted apoptosis corpuscles. Patient 3, for whom the disease had a duration of 50 days, like in the previous case, had obvious manifestations of diffuse alveolar damage. But the expression of the molecules of nucleocapsid SARS-CoV-2 and CD95 was weak positive.

The detection of the significant expression of SARS-CoV-2 antigens (nucleocapsid protein) in the epithelia of both large and small bronchi in the children who died on the 6th and 20th days of the disease is very interesting. A comparative evaluation of the data regarding SARS-CoV and MERS-CoV demonstrated that the extensive detection of SARS-CoV-2 antigens in the epithelial cells of the upper respiratory passages and the lung tissue in adult patients is unique among highly pathogenic coronaviruses [12,20]. It is important to underline the fact that in the case of PCR investigations of confirmed SARS-CoV-2 cases in adults, another SARS-CoV-2 antigen S-protein was also found in the epithelia of respiratory passages, according to the data regarding immunohistochemical investigation, while this protein was identified in smaller quantities in the endothelia and macrophages of lung tissue [21]. First of all, the results of this study specify the presence of a high viral load in the upper respiratory passages and a long-term, constant expression of the virus in severe COVID-19 cases [14,22,23]. Taking into account the literature data [24,25], the epithelia of the respiratory passages are, probably, the primary center of virus replication after air-droplet infection with the following viremia. With the exception of the above data, this can be confirmed based on the following results obtained by us: (1) the identification of SARS-CoV-2 in the tracheal material and the main bronchi, as well as the tissue of both lungs during the postmortem PCR investigation in children and (2) a significant expression of SARS-CoV-2 antigens in the endothelia of capillaries and larger vessels in those same cases. The data correspond to the results of other studies, which demonstrated the presence of the virus and SARS-CoV-2 antigens in the endothelial cells [26,27,28]. The data obtained here allow us to consider the epithelial and endothelial cells as the target cells of pathogenic influence [25,29,30]. But some authors suppose that the SARS-CoV-2 infection of endothelial cells is doubtful [31,32,33,34]. According to our study, in the patient who died on the 50th day of the disease, the virus was found in the material taken from the oropharynx and the nose, as well as in the broncho-alveolar lavage (according to PCR recorded up to day 47 of the disease), but the virus could not be identified during the postmortem investigation of the material taken from the trachea and lungs. These results correlate with immunohistochemical investigations. Therefore, we could not see the expression of SARS-CoV-2 antigens in the endothelia and bronchi. We could only identify “old” foci of SARS-CoV-2 antigen expression, presumably in the dead macrophages. According to the literature data, spike protein expression in the macrophages was noticed in the lungs of a 2-month-old patient who died due to this infection [35], and the expression of the SARS-CoV-2 S-protein was identified in the macrophages of the lungs of four adults [21]. The newly presented results demonstrate, first of all, that the virus’ presence in the macrophages of the lungs within different periods of the infectious process is a characteristic sign of COVID-19. Secondly, these results allow us to speak about the long-term presence of SARS-CoV-2 in macrophages, with the possibility of its persistence in a child’s tissue following reactivation. According to the results found by other researchers [36], the role of the macrophages in the studied cases probably had a double character, consisting of either a restriction of alveolar damage or, on the contrary, its expansion. To clarify this finding, further studies are needed. As can be seen from the presented material, a common trait of all these cases is damage to the endothelia in vessels of different sizes. According to the results of both a histological investigation and a more precise immunohistochemical investigation with the use of the antibodies of CD31, increasingly destructive damage to the endothelia of the capillaries of the interalveolar septa, arterioles, and venules was demonstrated during prolonged cases of COVID-19. An important aspect is the discussion of the causes of endothelium involvement in the infection process. Endothelia are found along the pathway of the virus’ distribution in organ tissues. The endothelial cells have an ACE2 receptor. These conditions allow the virus to adhere to and penetrate into the cells [29,37]. It has been found that the replication of coronaviruses takes place in the cell cytoplasm and does not cause any cytopathic changes [38,39]. A direct cytopathic effect of the viruses is not considered to occur on the epithelia of the bronchi or the endothelial cells. These cells should be considered the replication areas [40,41] but not as the targets for the cytopathic effects of the virus. The feedback cytokine response, capable of having a destructive effect or stimulating a similar effect in the infected cells [42], is significant. We found a significant expression of the apoptosis marker CD95 in the bronchial epithelia, the endothelia of vessels of different sizes, and the macrophages in Patients 1 and 2 (identified on the 6th and 20th days of the disease, respectively). Moreover, a positive reaction with antibodies of CD95 was observed in the apoptosis corpuscles, which simplified their identification. In contrast to Patient 1, apoptosis corpuscles in Patient 2 were identified in a more considerable number and were found everywhere in the bronchi, the direction of the capillaries in the interalveolar septa, the lumen of the alveoli, and the cytoplasm of the macrophages. We found that the localizations of the SARS-CoV-2 neocapsid protein, the apoptosis marker CD95, and the apoptosis corpuscles in different structures of the lungs probably are fundamentally significant in terms of understanding COVID-19 pathogenesis.

The data allowed us to identify apoptosis as a consequence of virus–host interactions. It is likely that the development of apoptosis has a significant influence on the virus–host interaction in the course of a new coronavirus infection [19], as it is one of the leading mechanisms of cell destruction in COVID-19-related deaths in children.

## 4. Materials and Methods

We analyzed reports of the autopsies and autopsy materials of 3 children who died due to COVID-19 during the period ranging from May 2021 to March 2022. Ethical clearance was obtained from the Local Ethics Committee of the St. Petersburg State Pediatric Medical University, under protocol number 30/06, on 27 September 2023. The disease was registered based on clinical manifestations and a positive SARS-CoV-2 PCR test. The clinical and laboratory features of the cases analyzed are presented in Table 1. SARS-CoV-2 PCR investigation was posthumously performed in trachea material and tissue from both lungs obtained during a pathological–anatomical autopsy. Paraffin sections were stained using hematoxylin and eosin, as well as Masson’s trichrome. An immunohistochemical investigation was performed using polyclonal antibodies of nucleocapsid SARS-CoV-2 (manufactured by GeneTex, Irvine, CA, USA); antibodies of CD20 and monoclonal antibodies of CD8+ (144B clone, manufactured by Diagnostic BioSystems, Pleasanton, CA, USA); monoclonal antibodies of CD31 (Jc/70A clone, manufactured by Diagnostic BioSystems, Pleasanton, CA, USA); and polyclonal antibodies of CD95 (manufactured by Diagnostic BioSystems, Pleasanton, CA, USA). The Mouse/Rabbit UnoVue HRP/DAB Detection System (manufactured by Diagnostic BioSystems, Pleasanton, CA, USA) was used for the IHC procedure. IHC staining was carried out according to both the recommendations of the manufacturers and our own experience [43,44,45].

CD95 is one of the most informative apoptosis markers. It is also called Fas, Apo-1, and TNFRSF6, as it is a member of the TNF 6 superfamily of receptors. It promotes apoptosis pathways while connected to the Fas ligand (FasL, CD178). It can also activate caspase-8 and caspase-3, finally leading to cell apoptosis [46]. It has been demonstrated that caspase-8 controls apoptosis, necroptosis, and pyroptosis and prevents tissue damage [47]. Capsase-8 activation plays a key role in apoptosis caused by SARS-CoV-2 [48], which highlights the importance of studying CD95 molecules in the lungs to evaluate the interactions between the virus and the host in cases of COVID-19. The autopsy material of the lung tissue from a child who died because of congenital heart defects and had no clinical symptoms of lung damage was selected as a control. Paraffin sections were stained using antibodies to CD95 and CD31 (Appendix A).

The results of the immunohistochemical staining stage were assessed semi-quantitatively and divided into several categories: acute expression (+++), moderate expression (++), and the presence of dark-brown granules in some cells (up to 10 cells in the field of vision under x×400 magnification)—slight expression (+). The assessment of the expression of the specified markers was carried out in up to 10 fields of vision under ×400 magnification.

## 5. Conclusions

According to the obtained data and the literature analysis, the vascular factor can be considered the key factor in COVID-19 pathogenesis. According to Tsankov [26], COVID-19 should be considered “an angiocentric disease”. The endothelium of microcirculation vessels is the primary and predominating target of the pathogenic influence of various factors. This fact can be confirmed by the presence of SARS-CoV-2 antigens (nucleocapsid protein) as a result of the penetration and replication of the virus in these cells and the further destruction of the infected endothelium cells due to induced apoptosis. Taking into account the unique role of the endothelium [40,49], its destruction and subsequent microcirculation disorder are the central points of the development of a capillary–alveolar block, tissue hypoxia, and disseminated intravascular coagulation syndrome (DIC-syndrome) [45,50,51], leading to the development of multiple organ dysfunction syndrome (MODS) and, ultimately, death.

The presented results of this study have significant value regarding the development of endothelium investigation methods, the production and usage of new endothelium-protective therapy in COVID-19 medical practice, and the prevention of other diseases, even those of non-infectious origin [44].

## Figures and Tables

**Figure 1 ijms-24-16750-f001:**
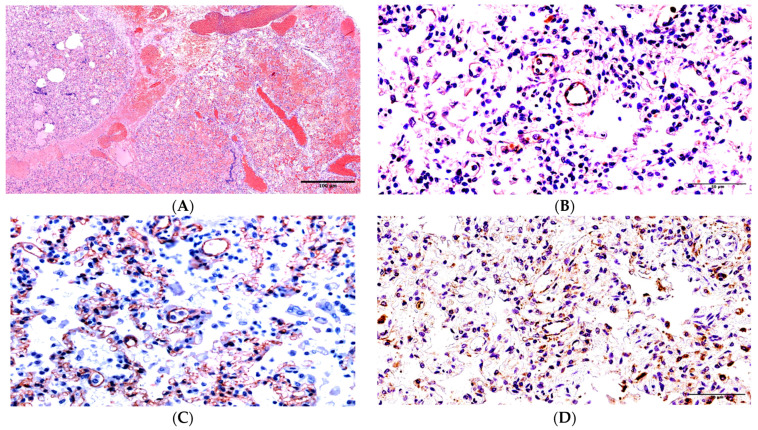
Changes in the lungs in Patient 1. (**A**) Widened vessels, hemorrhages, dystelectasis, and serous-hemorrhagic exudate in the alveoli lumina. H&E under ×100 magnification. (**B**) SARS-CoV-2 nucleocapsid expression on the walls of the micro-vessels and in interalveolar septa under ×400 magnification. (**C**) CD31 immunostaining, the irregular capillary network, and the desquamation of some endotheliocytes under ×400 magnification. (**D**) The widespread expression of apoptosis marker CD95 on the walls of micro-vessels and the interalveolar septa under ×400 magnification.

**Figure 2 ijms-24-16750-f002:**
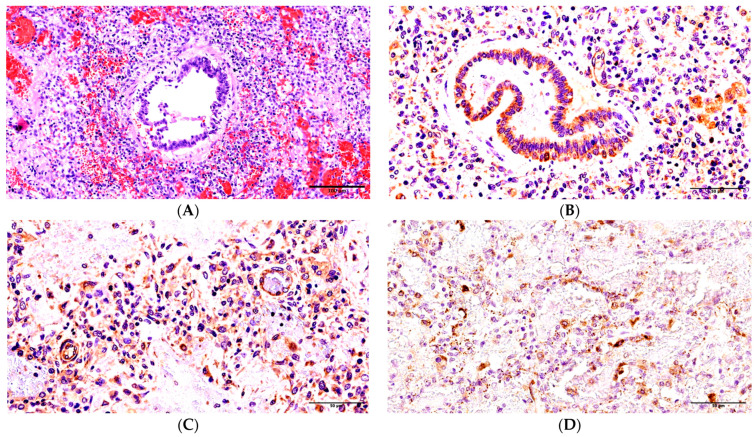
Changes in the lungs in Patient 2. (**A**) The serous-fibrinous exudate in the alveoli lumina, hemorrhages, the dissection of the bronchus wall, and inflammatory lymphocytic infiltration were noted via H&E under ×200 magnification. (**B**) The diffuse expression of SARS-CoV-2 nucleocapsid in the bronchus epithelium and the acute expression of the antigen in macrophages were noted under ×400 magnification. (**C**) The significant expression of SARS-CoV-2 nucleocapsid in the walls of the micro-vessels, the interalveolar septa, and many apoptosis corpuscles was noted under ×400 magnification. (**D**) The significant expression of the apoptosis marker in the walls of the micro-vessels, the interalveolar septa, and many apoptosis corpuscles (insertion) was noted. CD95 staining under ×400 magnification.

**Figure 3 ijms-24-16750-f003:**
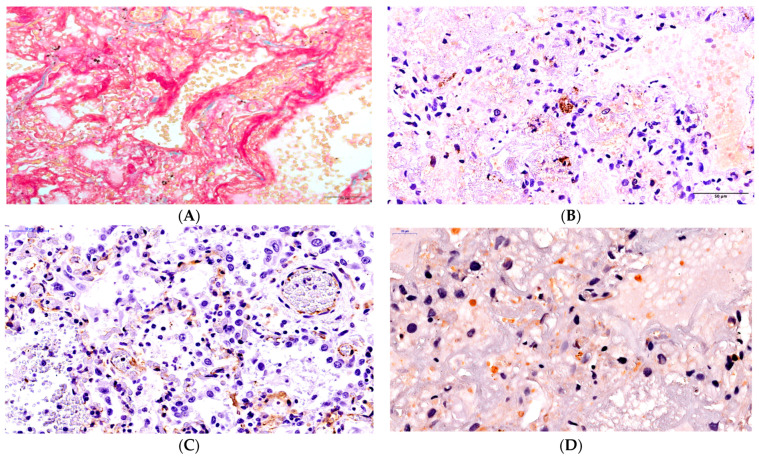
Changes in the lungs in Patient 3. (**A**) Fibrin fibers in alveoli lumina. Staining via trichrome according to Masson’s method under ×400 magnification. (**B**) Some foci of the virus antigen expression under ×400 magnification. (**C**) Damage to the integrity of the walls of the micro-vessels and visible adelphomorphous fragments of the endothelial cells of the capillary network of the lung. CD31 staining under ×400 magnification. (**D**) The weak positive/doubtful expression of the apoptosis marker in the apoptosis particles and the desquamated bronchial epithelium. CD95 staining under ×600 magnification.

**Table 1 ijms-24-16750-t001:** Clinical and laboratory features of COVID-19 cases.

Patient	Age	Duration of the Disease	Comorbid Conditions	Results of Laboratory Investigations		
				Ferritin (mkg/L) *n* = 15.00–120.00	D-Dimer (ng/mL) *n* = 0–250.0	C-Reactive Protein (mg/L) *n* = 0–5.80
1. Girl E.	1 year, 7 months	6 days	Acute myeloblastic leukemia, M4 variant with eosinophilia, CD19 co-expression, CBFB-inv [16] gene reorganization, and neuroleucosis	1654.7 (2 days of the disease) 3775.1 (4–5 days of the disease)	4158.0 (1 day of the disease) 5160.0 (4–5 days of the disease)	157.8 (1 day of the disease) 277.5 (2 days of the disease) 226.3 (4–5 days of the disease)
2. Boy M.	1 year, 5 months	20 days	Hydrocephaly	714.60 (16 days of the disease) 391.4 (20 days of the disease)	81.0 (15 days of the disease) 1174.0 (19 days of the disease) 11009.0 (20 days of the disease)	9.6 (13 days of the disease) 69.30 (18 days of the disease) 59.1 (20 days of the disease)
3. Boy X.	12 years, 7 months	50 days	Acute lymphoblastic, leukemia, B II immunological variant, and condition after bone marrow transplantation	3091.1 (1 week of the disease) 3840.7 (18–26 days of the disease) 8947.1 (29–30 days of the disease) 10,899.3 (44 days of the disease)	387.0 (29–30 days of the disease) 2292.0 (44 days of the disease)	27.0 (18–26 days of the disease) 45.1 (29–30 days of the disease) 298.0 (31–37 days of the disease) >320.0 (44 days of the disease)

## Data Availability

Data is contained within the article and Appendix A Not applicable.

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
