# Peer review of "COVID-19 in Children: Molecular Profile and Pathological Features"

_ijms, 2023, doi:10.3390/ijms242316750_

Round 1

Reviewer 1 Report

Comments and Suggestions for Authors

In this study, the authors have studied autopsy lung material of 3 children that have died due to COVID-19.

The study involved histopathological and immunohistochemical analysis, as well as some laboratory data.

- A major issue here is English language, that needs extensive editing. In many cases it is difficult to understand what is written, for example in lines 57, 148-149, 159-161, 242. Grammatical errors occur very often, for example in lines 19, 36, 112, 234, 257, 262-263.

- The authors state that ‘the data presented by us allow to identify apoptosis as a consequence of the interaction between virus-host’. I cannot find how this is shown. The only result relative to apoptosis is CD95 staining (single and not double staining!), which is a marker of acute lymphoblastic leukemia cells.

- The authors state that the common finding of the 3 cases is the involvement of endothelium. I cannot see this, either. CD31 immunohistochemical staining is seen only in Figure 1C (patient 1), whereas in patient 2 CD31 staining is not shown and in patient 3 endothelial cells were not identified. Moreover, in Figures 1C and 3C the cyan/blue levels are very high.

- How is it explained that in patient 3 (whose comorbidity is acute lymphoblastic leukemia), there is no positive staining for CD95?

- Presentation of images is not satisfying. Panels should have same size and, in some cases, closer images are needed.    

- Materials and Methods section needs expansion, with separate subsections (i.e., study design, immunohistochemistry procedure in detail etc).  

Comments on the Quality of English Language

English language needs extensive editing

Reviewer 2 Report

Comments and Suggestions for Authors

The MS Covid-19 in children: molecular profile and pathological 2

featuresis a very interesting research paper.  . They present the results of the study of

the lungs of 3 children died due to COVID-19 with a predominant involvement of their respiratory organs at different stages of the disease (5, 21, and 50 days)

There are some details that need to be revised

Comment 1: Abstract is informative, rather well written, however some details (lines 26-29) may be omitted or shortened: The received results of the study could be useful for the development and usage of endothelium-protective therapy to prevent the progress of disseminated intravascular coagulation syndrome and multiple organ dysfunction syndrome in medical practice associated with COVID-19 and other diseases, even of noninfectious origin.

Comment 2: Introduction section is generally rather well written, however aim of the study should be added at the end of the Introduction section

Comment 3: Authors should revise:

Line 165:  Expression of apoptosis marker CD95 noticed in some endotheliocytes of arterioles.

Instead there should be: endothelial cells of arterioles

Comment 4: Results section is generally rather well written

Comment 5 Discussion section is generally rather well written.

Fer sentences should be explained/revised:

Comment 6: Tape-like fibrin masses covering alveoli in the form of “hyaline membranes”

Instead: Tape-like fibrin masses covering alveoli in the form of so called hyaline membranes. Explain what they are

Comment 7: Line 165: Expression of apoptosis marker CD95 noticed in some endotheliocytes of arterioles.

Did you have maybe the results of staining in some control group (either healthy subject or some other disorder). Expression of apoptosis marker CD95 might be noticed also in healthy subjects

Comment 8: Line 186: Opposite the previous case there was a significant T-limphocytic

inflammatory infiltration

Instead: Opposite the previous case there was a significant T-lymphocytic

inflammatory infiltration

Comment 9: Line 242: We do not consider a direct cytopathic effect of the viruses on the epithelium of bronchi, endothelial cells.

What do you mean by: “cytopathic effect”

Comment 10 Language should be checked by the native speaker.

Comments on the Quality of English Language

Comment 10 Language should be checked by the native speaker.

Round 2

Reviewer 1 Report

Comments and Suggestions for Authors

The authors have made some changes in the manuscript, but I still believe that the data provided here are not strong enough to support publication.

As I said from the first round, the study involves mainly histopathological and immunohistochemical analysis; for this reason, findings presented should have scientific soundness and be reliable. Unfortunately, I think this is not the case.

- The authors present results from immunostainings and histopathological analysis, but the figures have not the quality expected.  More specifically, figures 1C, 3C and 3D are examples of not high-quality microphotographs.

Moreover, data would be more complete if for all three patient cases there were figures of unstained control, H&E, Masson, SARS-CoV-2 Ag, CD31 and CD95 (in patient 2 CD31 staining is still not shown). CD4, CD8, CD20 should be also shown, since they are mentioned in Results (lines 64, 96) and in Methods (lines 262-263).

Furthermore, authors include in their results findings that, in my opinion, are not supported by the figures. These are a) line 75, ‘CD31 staining shows fragmentation of capillary network’. I cannot see how this is shown in Figure 1C. b) line 78, ‘acute expression of CD95 in macrophages’. From Figure 1D macrophages cannot be identified. c) line 112, ‘CD95 expression in the apoptosis particles’. How are these particles shown? Figure 2D does not allow identification of microvessel walls, either.

- I cannot understand the addition of control in Figure 4. Staining for CD95 is negative (Figure 4A is black and white!!) and staining for CD31 is positive, but it does not link with the rest results. And clearly, Figure 4 should not be placed in the Materials and Methods section.

Comments on the Quality of English Language

English language needs moderate editing

Round 3

Reviewer 1 Report

Comments and Suggestions for Authors

No further comments.